# Short-Term Continuous Positive Air Pressure Treatment: Effects on Quality of Life and Sleep in Patients with Obstructive Sleep Apnea

**DOI:** 10.3390/medicina58030350

**Published:** 2022-02-25

**Authors:** Laima Kondratavičienė, Kęstutis Malakauskas, Guoda Vaitukaitienė, Tomas Balsevičius, Marius Žemaitis, Skaidrius Miliauskas

**Affiliations:** 1Department of Pulmonology, Lithuanian University of Health Sciences, 44307 Kaunas, Lithuania; kestutis.malakauskas@lsmu.lt (K.M.); guoda.vaitukaitiene@lsmu.lt (G.V.); marius.zemaitis@lsmu.lt (M.Ž.); skaidrius.miliauskas@lsmuni.lt (S.M.); 2Department of Otorhinolaryngology, Lithuanian University of Health Sciences, 44307 Kaunas, Lithuania; tomas.balsevicius@lsmu.lt

**Keywords:** continuous positive air pressure, Epworth Sleepiness Scale, obstructive sleep apnea, Pittsburg Sleep Quality Index, quality of life, 36-Item Short Form Health Survey

## Abstract

*Background and Objectives*: The aim of this study was to evaluate short-term continuous positive air pressure (CPAP) treatment for health-related quality of life (HRQL) in patients with obstructive sleep apnea. *Materials and Methods*: Our subjects were 18–65 years old, diagnosed with moderate-to-severe obstructive sleep apnea and treated with CPAP between January 2020 and June 2021 in Hospital of Lithuanian University of Health Sciences Kaunas clinics. All the patients completed the Epworth Sleepiness Scale (ESS), the 36-Item Short Form Health Survey (SF-36), the and Pittsburgh Sleep Quality Index (PSQI) before and after 3 months of treatment. Polysomnography was also repeated. Statistical analyses were performed using SPSS 27.0 software. The value of *p* < 0.05 was considered as statistically significant. *Results*: The active-treatment group comprised 17 subjects with a mean age of 51.9 ± 8.9 years. The total SF-36 questionnaire score improved from 499.8 ± 122.3 to 589.6 ± 124.7 (*p* = 0.012). The SF-36 role limitations due to emotional problems (*p* = 0.021), energy (fatigue) (*p* = 0.035), and general health (*p* = 0.042) domains score significantly improved after CPAP treatment for 3 months. The PSQI mean score at baseline was 12.6 ± 2.9 and in the post-treatment group, it was −5.5 ± 2.3 (*p* = 0.001). The ESS also changed significantly from a pretreatment mean score of 10.9 ± 5.7 to −5.3 ± 3.2 (*p* = 0.002) after 3 months. *Conclusions*: Improvement in HRQL is seen even after a short treatment period with CPAP. Questionnaires are a good tool to evaluate CPAP treatment efficacy.

## 1. Introduction

Obstructive sleep apnea (OSA) is a common health problem, affecting individuals’ health-related quality of life (HRQL) as well as sleep. It is described as a chronic disease with repeatable respiratory pauses during sleep, accompanied by episodic hypoxia and sleep fragmentation [1]. There are no exact data for the prevalence of OSA in the Lithuanian population. Various studies show different prevalence of OSA, but according to the Wisconsin Sleep Cohort Study, 4% of middle-aged men and 2% of middle-aged women have OSA [2], and in other studies, OSA was diagnosed in 37% of men and 50% of women in a 5-year period [3]. Continuous positive air pressure (CPAP) therapy is the standard treatment of choice for moderate or severe OSA [4]. For patients on CPAP, it is very important to monitor care and observe the effect of treatment to ensure its efficacy. Questionnaires are simple and low-effort tools to evaluate quality of life and sleep. In this study, we compared the effect of short-term CPAP treatment on quality of life and sleep by using three different questionnaires: the Epworth Sleepiness Scale (ESS), the 36-Item Short Form Health Survey (SF-36), and the Pittsburgh Sleep Quality Index (PSQI). CPAP devices were not previously used by these patients and this treatment method for all subjects, diagnosed with OSA, was new.

## 2. Materials and Methods

### 2.1. Subjects

This was a longitudinal study, evaluating the quality of life and sleep in patients with OSA before treatment (when OSA was diagnosed) and after 3 months of CPAP therapy.

The 34 subjects, diagnosed with moderate-to-severe OSA, were enrolled in the study from January 2020 through June 2021. Seventeen patients underwent the final investigation after 3 months of treatment with CPAP. The main reasons for drop-out were CPAP intolerance, the coronavirus disease (COVID-19) pandemic, and financial reasons. In Lithuania, CPAP devices are not reimbursed by the government.

The study was approved by the Kaunas Regional Biomedical Research Ethics Committee (no. BE-2-23, 19 May 2020, Kaunas, Lithuania).

The inclusion criteria were: age 18–65 years old, diagnosis of moderate-to-severe OSA, and signed informed consent form.

The exclusion criteria were: children up to 18 years old and adults older than 65 years, significant mental and/or internal organ disease that might affect the study protocol, absence of a signed informed consent form, and the judgment of the investigator.

All subjects underwent detailed clinical investigation, which included recording complaints, medical and surgical history, and comorbidities, as well as performing a complete physical examination. Patients were seen in the Hospital of Lithuanian University of Health Sciences (LUHS) Kaunas Clinics Outpatient Clinic, and they underwent consultation by an otorhinolaryngologist to rule out any nasal pathology (e.g., nasal polyposis, deviated nasal septum, insufficiency of the nasal valve). During the first visit, complaints were registered and subjective daytime sleepiness was evaluated by the ESS.

Those who were suspected of having OSA were referred to the Sleep Laboratory in the LUHS Kaunas Clinics Pulmonology Department for overnight diagnostic polysomnography. This study was performed using the Alice 6 LDx diagnostic sleep system (Philips Respironics, Murrysville, PA, USA). The absence of airflow for more than 10 s was defined as apnea, and a decrease in airflow for at least 10 s accompanied by a 3% reduction in SpO_2_ or arousal as hypopnea. The apnea–hypopnea index (AHI) was recorded per hour of sleep during the study. Patients with OSA were classified according to the AHI: mild OSA was defined as AHI ≥ 5 but <15, moderate OSA as AHI ≥ 15 but <30, and severe OSA as AHI ≥ 30 [5,6].

After OSA diagnosis, each patient spent another night in the sleep laboratory for CPAP titration to select the optimal pressure at which the CPAP device eliminated abnormal breathing events. Patients diagnosed with moderate or severe OSA were invited to participate in the clinical study.

### 2.2. ESS

The ESS is a subjective method to evaluate daytime sleepiness [7]. It comprises eight self-administered questions describing the likelihood of falling asleep in various situations. The subject is asked to rate every situation on a scale from 0 to 3, which ranges from 0 (no sleepiness) to 24 (extremely sleepy). The higher the ESS score, the more severe the daytime sleepiness. ESS was evaluated at baseline (before the CPAP therapy) and repeated after 3 months of CPAP treatment. The Lithuanian version of the ESS was previously evaluated for its internal consistency and was used in a number of studies [8,9,10].

### 2.3. SF-36 and PSQI

The SF-36 is a tool to evaluate HRQL [9]. This questionnaire consists of 36 questions and covers eight domains of health: physical functioning, role limitations due to physical health, role limitations due to emotional problems, energy (fatigue), emotional well-being, social functioning, pain, and general health. Scores range from 0 (minimum welfare) to 100 (maximum welfare). Physical functioning, role limitations due to physical health, pain, and general health domains were combined as one physical component score, and the domains of role limitations due to emotional problems, energy (fatigue), emotional well-being, and social functioning domains comprised the mental component score. The Lithuanian-validated version of the SF-36 questionnaire was used [11].

The Lithuanian version of the PSQI, also a questionnaire, relates to sleep habits during the previous month [8]. It consists of 19 questions that are self-related and 5 questions that are rated by the bed partner, if there is one. Only self-related questions are analyzed. They are combined to form a seven “component” scores from 0 to 3 points, with a total questionnaire score ranging from 0 to 21 points. A PSQI ≤ 5 is defined as no disorder, PQSI > 5 and ≤10 as an episodic sleep disorder, PQSI > 10 and ≤15 as a moderate sleep disorder, and PQSI > 15 and ≤21 as a severe sleep disorder [12].

### 2.4. Interventions and Follow-Up

Study subjects were asked to return for a follow-up visit 3 months after treatment began. CPAP treatment compliance was checked by downloading data from the CPAP device. We defined compliance with treatment as use of the CPAP device for more than 4 h per night, for more than 70 percent of nights, and AHI < 5.

Clinical study subjects filled out PSQI and SF-36 questionnaires and rated ESS at the baseline and after 3 months of CPAP treatment. The patients also underwent control polysomnography.

### 2.5. Statistical Analysis

We analyzed two groups of patients: those enrolled at baseline (pretreatment group) and those with continuous CPAP treatment (posttreatment group). Quantitative variables were described using the mean and standard deviation or median at 25th–75th percentiles. To describe the distribution and changes in all quantitative variables, the nonparametric the Mann–Whitney U-test for two dependent samples was used.

Changes in SF-36 domain scores, PSQI, and ESS were calculated by subtracting posttreatment values from pretreatment values. The connection between each domain and the other variables was calculated using the Spearman correlation analysis.

IBM SPSS Statistics for Windows, version 27.0 (IBM Corp., Armonk, NY, USA), was used for statistical analysis. Statistical significance was considered to be *p* < 0.05.

## 3. Results

A total of 34 patients were enrolled in the study: 29 (85.3%) were men and 5 (14.7%) were women. Patients that dropped out did not differ from the remaining study population in any other way. At baseline, according to chi-squared and Fisher’s exact tests, the pretreatment group was 14 (82.4%) men and 3 (17.6%) women. After 3 months of treatment with CPAP, the results were 15 (88.2%) men and 2 (11.8%) women (*p* = 1.0). According to the pretreatment and posttreatment group results, sex, age, body mass index (BMI), SF-36 domains and PSQI scores, perception of ESS, and AHI did not differ significantly (Table 1).

For the pretreatment and posttreatment groups (subjects who continued treatment with CPAP and follow-up was performed), the BMI and most polysomnography data did not change significantly (Table 2).

After 3 months of CPAP treatment, we observed a significant improvement in the HRQL scores of the patients. The total SF-36 score improved from 499.8 ± 122.3 to 589.6 ± 124.7 (*p* = 0.012). The SF-36 role limitation scores due to emotional problems (*p* = 0.021), energy (fatigue) (*p* = 0.035), and general health (*p* = 0.042) domains significantly improved after 3 months of CPAP treatment. Combining the SF-36 domains into physical and mental component scores, we found significant improvement in both component scores in the posttreatment group (*p* = 0.044 and *p* = 0.009, respectively) (Table 3).

The sleep quality evaluation using the PSQI questionnaire also changed significantly after 3 months of CPAP treatment: at baseline, the mean evaluation was 12.6 ± 2.9, and in the posttreatment group, it was −5.5 ± 2.3 (*p* = 0.001). After short-term treatment with CPAP, even 53.3% of subjects rated their sleep quality as not having any sleep disorders; none of pretreatment results were in that category.

The perception of sleepiness, when evaluating the ESS during the CPAP treatment period, also significantly changed: the pretreatment ESS mean score was 10.9 ± 5.7; after 3 months it was −5.3 ± 3.2 (*p* = 0.002) (Table 4).

According to the nonparametric Kruskal–Wallis one-way analysis of variance, we found significant PSQI and SF-36 mental component score distribution (even though the SF-36 questionnaire domains did not change individually, the physical component score was *p* = 0.133, and the mental component score was *p* = 0.047).

According to the Spearman correlative analysis, the ESS score changes had significant direct correlation with SF-36 (role limitations due to emotional problems domain (r = 0.631, *p* = 0.012) and the social functioning domain (r = 0.54, *p* = 0.038)) (Figure 1).

The ESS score also featured a significant improvement in the SF-36 mental component score (r = 0.676, *p* = 0.06), but not in the physical component score (r = −0.111, *p* = 0.693) (Figure 2).

## 4. Discussion

In this study, we analyzed the short-term CPAP treatment effect on HRQL, quality of sleep, and perception of sleepiness for patients with moderate-to-severe OSA by using three different tools: the SF-36, PSQI, and ESS. Various studies have used many different tools for the evaluation of quality of life for patients with OSA because there is no one universal instrument that fits every patient’s requirements. The majority of current studies investigate long-term CPAP effects; few evaluate the effects of short-term CPAP treatment on the quality of life. Furthermore, quality of life has been assessed by several different instruments. It is very important to evaluate the early results of treatment to understand the course of a disease, to solve obstacles that patients encounter, and to propose other treatment options. These steps are needed to ensure that patients are not left alone in their fight with a disease.

To comprehensively evaluate the mental changes as well as the physical changes in the quality of life of CPAP patients, different HRQL questionnaires were employed. We found that CPAP improved patients’ mental health in the short term. However, we did not observe any improvement in patients’ physical health after 3 months of CPAP therapy. This finding can be related to the fact that CPAP usually normalizes breathing during sleep and mainly reduces symptoms (e.g., snoring, daytime sleepiness, vigilance). In addition, the BMI of our study subjects in the 3-month period did not change significantly; therefore, no improvement in the physical component of the quality of life score was observed. To evaluate sleepiness, we used the PQSI and ESS, which significantly changed after 3 months with CPAP treatment, thereby demonstrating treatment benefits to study group patients.

According to our study results, we found that short-term CPAP treatment changes the quality of life but not sleep structure, as the main polysomnography data did not change significantly.

### 4.1. SF-36

SF-36 is a generic questionnaire designed to measure the eight most important domains of life, but it is less specific in the evaluation of sleep quality and might not correlate with polysomnography data. Various clinical trials proved that even though the SF-36 is not a disease-specific questionnaire for OSA, it is used in the evaluation of quality of life [13]. Berg et al., in a randomized clinical trial, compared CPAP and a twin block mandibular advancement splint (MAS) treatment effect on quality of life in patients with non-severe OSA during a 12-month period [14]. Even though this study’s main research focus was non-severe OSA, and CPAP treatment was continued for a much longer period, it showed that CPAP led to a significant improvement in the SF-36 vitality domain. This was proven in our study (vitality is comparable due to the physical health domain and the physical component score). This might be associated with the different BMIs of the subjects and the differing levels of OSA severity.

The effect of long-term CPAP on quality of life was also investigated by Pitchel et al. [15]. In their 6-month CPAP treatment group, only the SF-36 vitality (energy) domain improved, but after 12 months of treatment, five domains of the SF-36 improved: physical functioning, role limitation due to physical health, social functioning, vitality, and general health. Compared with our research, in which we found significant improvement in three SF-36 domains after 3 months, we can conclude that even short-term CPAP treatment changes patients’ quality of life.

Silva et al. compared three different tools for quality of life for patients with OSA and their correlations without CPAP treatment. The SF-36 questionnaire was one tool [16]. The main finding and aim of this study was that correlations between various quality-of-life evaluation tools are not high and might cover different domains of life. Therefore, the decision over which tool to choose is taken by the investigator. Analyzing our data and results and comparing them with other researchers’ results, the SF-36 questionnaire is a useful instrument covering multiple aspects of a subject’s life.

### 4.2. PSQI and ESS

The PSQI and ESS are two different questionnaires, assessing different situations and daytime somnolence: PQSI describes quality of sleep at a night and ESS covers the perception of sleepiness in everyday situations. Buysse et al. characterized these two different tools and their relationship [17]. The main conclusion was that they are not perfect tools for polysomnographic sleep disorder screening. However, this study did not cover the treatment aspect of OSA. According to our study and findings, we strongly argue that both the PSQI and ESS could reveal the effects of CPAP treatment. In our study, a significant correlation was found between the evaluation of the PSQI and SF-36 mental component scores after a 3-month treatment period. Consequent changes were found in the same areas of the questionnaires, as would be expected when comparing specific and general tools. These findings could support the conclusion that better sleep quality is associated with a patient’s improved psychological condition.

A detailed and comprehensive meta-analysis by Mollayeva et al. analyzed the PSQI as a screening tool for sleep disorders and suggested that the PSQI is the only standardized clinical tool that covers sleep quality at nighttime [18]. According to our study data, we agree with this proposition; we showed that even after short-term OSA treatment with CPAP, more than half of the study subjects rated their sleep quality as featuring no sleep disorders, in contrast to their pretreatment results.

The ESS is usually used as a first-choice instrument to detect sleepiness. Its usage is controversial as it has only limited OSA screening capability, and it does not show the severity of disease. Walker et al. analyzed the variability of the ESS at two time points: at a primary visit before the sleep study and after the performance of the sleep study [19]. The results showed that the ESS scores varied widely between different patients, but it is well suited for categorizing patients as sleepy or not sleepy. As was also noted in our study, the perception of sleepiness when scoring ESS does not correlate with AHI. The retrospective analysis of patients with sleeping disorders by Lee et al. also compared the variability of ESS scores at two time points [20]. They proposed that the ESS should not be used in clinical practice for assessing individual changes or prioritizing patients for diagnostic sleep disorder procedures. However, the ESS can be employed to evaluate compliance with the CPAP therapy, as was performed in our study and observed in the study by Economou et al. [21], despite different subjects’ perceptions of sleepiness.

This clinical trial has some limitations that should be highlighted. The main limitation of this study is the relatively small patient sample. Both groups of our study patients, those who did not start or who discontinued CPAP treatment and those who used the CPAP device and had good compliance, were similar to each other. We believe that the patients enrolled in the study could be representative of all patients with moderate or severe OSA. The main difficulty in this study was over the patients that did not start treatment because of financial obstacles (CPAP devices are not reimbursed by the government), or did not understand the severity of their disease in itself (in some cases, patients did not feel subjective somnolence). Another problem was that patients discontinued treatment because of inconvenience (e.g., “the CPAP device mask is not convenient to use”, “my partner could not sleep with me due to sounds CPAP device sent”, “unattractive”). Buyse et al. published an interesting narrative review regarding adherence to CPAP treatment, and the main finding was that adherence increased when reimbursement for the CPAP devices began, even though the control of CPAP device usage was tightened (data on CPAP device usage downloaded from CPAP device after 3 months) [22]. A further limitation of this study was that only questionnaires, which are very subjective and can be biased, were analysed. Since the subjects knew that they would be treated with CPAP devices, their responses after 3 months of treatment could be easily predicted.

## 5. Conclusions

Short-term CPAP treatment improves the mental aspects of the HRQL, the perception of sleepiness, and sleep quality in patients with moderate-to-severe OSA. The absence of changes in the physical health component strongly suggest that patients did not change their daily life routines. The PSQI and ESS are instruments that are well suited to the evaluation of compliance with CPAP treatment.

## Figures and Tables

**Figure 1 medicina-58-00350-f001:**
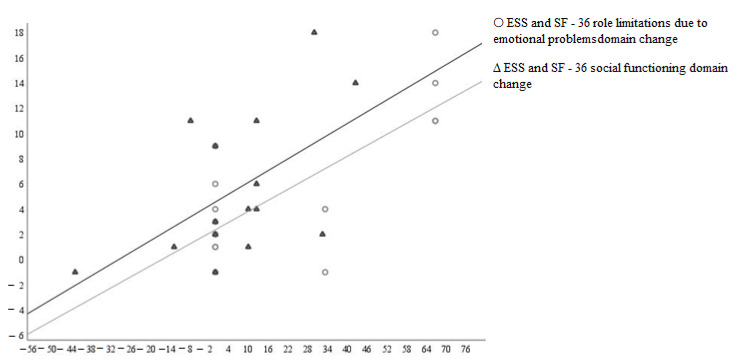
Scatter plot showing the relationship between changes in SF-36 role limitations due to emotional problems and social functioning domains and the Epworth Sleepiness Scale (ESS).

**Figure 2 medicina-58-00350-f002:**
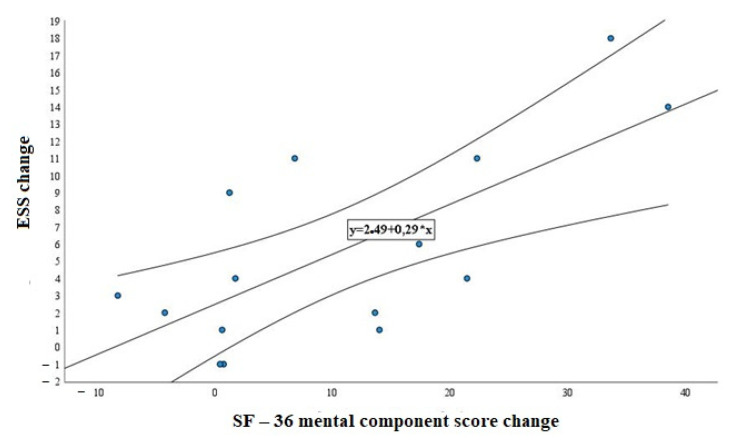
Scatter plot showing the relationship between improvements in the SF-36 mental component score with CPAP treatment and the Epworth Sleepiness Scale (ESS).

**Table 1 medicina-58-00350-t001:** Characteristics of patients who dropped out compared with remaining study population.

		Patients Who Dropped Out (*n* = 17)	Patients in Active Treatment Group (*n* = 17)	*p*-Value *
		Mean (Standard Deviation)		
Age, years		52.4 (9.1)	52.6 (9.0)	0.708
BMI, kg/m^2^		36.6 (4.3)	39.1 (6.0)	0.093
SF-36 domains	Physical functioning	67.6 (21.8)	67.3 (25.9)	0.614
	Limitations due to physical health	60.3 (34.3)	52.1 (40.0)	0.447
	Limitations due to emotional problems	60.8 (42.8)	54.9 (47.1)	0.292
	Energy (fatigue)	59.1 (17.1)	58.4 (16.2)	0.611
	Emotional well-being	62.3 (19.2)	67.2 (16.4)	0.497
	Social functioning	74.1 (25.2)	77.9 (21.8)	0.632
	Pain	84.3 (17.5)	68.8 (27.9)	0.01 *
	General health	50.1 (18.2)	51.2 (14.6)	0.8
ESS, points		8.9 (3.9)	10.2 (5.4)	0.161
PSQI, points		9.3 (3.4)	12.5 (2.9)	0.435
AHI, per h		57.7 (23.5)	76.1 (24.3)	0.980

******p*-value < 0.05, according to nonparametric Wilcoxon test. AHI: apnea hypopnea index, BMI: body mass index, CPAP: continuous positive airway pressure, ESS: Epworth Sleepiness Scale, PSQI: Pittsburgh Sleep Quality Index; SF-36: 36-item short form survey.

**Table 2 medicina-58-00350-t002:** Polysomnography data of subjects who continued treatment.

		Pretreatment (*n* = 17)	Posttreatment (*n* = 17)	*p*-Value
		Mean (Standard Deviation)		
PSG	AHI (/h)	76.1 (24.3)	67.7 (26.6)	n.s.
	Oxygen desaturation index (/h)	66.9 (23.6)	34.2 (31.9)	0.037 *
	Mean of SpO_2_ when awake (%)	91.4 (2.5)	92.7 (2.3)	n.s.
	Minimum SpO_2_ level during sleep (%)	67.1 (12.5)	74.5 (8.3)	n.s.
	Sleep efficiency (%)	77.0 (12.2)	82.5 (7.7)	n.s.
	Arousal index (/h)	71.5 (23.2)	71.4 (27.3)	n.s.
BMI (kg/m^2^)		39.23 (6.1)	38.3 (5.5)	n.s.

******p*-value < 0.05, according to nonparametric Wilcoxon test. AHI: apnea hypopnea index, BMI: body mass index, CPAP: continuous positive airway pressure, n.s.: not clinically significant (*p* > 0.05), PSG: polysomnography, SpO_2_: oxygen saturation by pulse oximeter.

**Table 3 medicina-58-00350-t003:** SF-36 domains at baseline (pretreatment) and after 3 months of CPAP treatment (posttreatment).

	Pretreatment (*n* = 17)	Posttreatment (*n* = 17)	Adjusted Difference (Mean, 95% CI)	*p*-Value
	Mean (Standard Deviation)			
Physical functioning	67.35 (25.98)	74.12 (22.93)	6.77 (−5.0–20.0)	0.12
Role limitations due to physical health	52.06 (40.04)	73.53 (35.87)	21.47 (0–25.0)	0.102
Role limitations due to emotional problems	54.9 (47.06)	86.28 (23.75)	31.38 (0–66.7)	0.012 *
Energy (fatigue)	58.41 (16.12)	66.76 (12.24)	8.35 (−5.0–20.0)	0.081 *
Emotional wellbeing	67.18 (16.39)	75.52 (13.03)	8.34 (−8–20.0)	0.238
Social functioning	77.39 (21.83)	86.59 (16.92)	9.2 (0–12.5)	0.163
Pain	68.82 (27.94)	72.38 (29.70)	3.56 (10.0–10.0)	0.347
General health	51.18 (14.63)	62.64 (17.06)	11.46 (0–20.0)	0.002 *
Physical component score	60.01 (21.87)	70.67 (22.48)	10.66 (6.25–11.8)	0.058
Mental component score	64.47 (21.49)	78.78 (14.06)	11.31 (0.6–21.4)	0.009 *

* *p*-value < 0.05, according to nonparametric Wilcoxon test. CPAP: continuous positive airway pressure.

**Table 4 medicina-58-00350-t004:** PSQI and ESS score at baseline (pretreatment) and after 3 months of CPAP treatment (posttreatment).

PSQI	Pretreatment (*n* = 17)	Posttreatment (*n* = 17)		
	Frequency (%)			
PSQI ≤ 5	0	53.3		
5 < PQSI ≤ 10	26.7	40		
10 < PQSI ≤ 15	46.7	6.7		
15 < PQSI ≤ 21	26.7	0		
	**Mean (Standard Deviation)**		**Adjusted Difference (95% CI)**	***p*-Value**
PSQI general score	12.6 (2.9)	5.5 (2.3)	7.7 (2.9)	0.001 *
ESS	10.9 (5.7)	5.3 (3.2)	5.6 (5.7)	0.002 *

* *p*-value < 0.05, according to nonparametric Wilcoxon test. CPAP: continuous positive airway pressure, ESS: Epworth sleepiness scale, PSQI: Pittsburgh sleep quality index. PSQI ≤ 5: no sleep disorder, 5 < PSQI ≤ 10: episodic sleep disorder, 10 < PSQI ≤ 15: moderate sleep disorder, 15 < PQSI ≤ 21: severe sleep disorder.

## Data Availability

Not applicable.

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
