# Peer review of "Short-Term Continuous Positive Air Pressure Treatment: Effects on Quality of Life and Sleep in Patients with Obstructive Sleep Apnea"

_medicina, 2022, doi:10.3390/medicina58030350_

Round 1
Reviewer 1 Report
In this manuscript, Kondratavičienė and colaborators aimed to highlight how a short-term CPAP regimen might influence sleep and quality of life in patients with obstructive sleep apnea. The article is well written and the use of a multi-questionnaire assessment represents an interesting approach for the evaluation of CPAP’s efficiency. However, there are several aspects to be further detailed and improved:
- The methodology lacks a more detailed description of CPAP device (type, selected ventilation parameters). It would be interesting to correlate these parameters with the questionnaires scores.
- I highly recommend the authors to perform a ROC analysis in order to more specifically evaluate how a poor baseline score can predict the adherence to CPAP and, conversely, how CPAP can predict an improvement regarding quality of life (as expressed by questionnaires' scores).
- It would be interesting to correlate the questionnaire scores with the patients’ performance at cardiopulmonary exercise testing (eg. VO2 max), which represent an increasingly used tool for quality of life assessment in CPAP patients (of course, only if some of the enrolled subjects underwent this test).
- The small data sample represents a major drawback.
The English language needs to be assessed by a native speaker, as several phrases are oddly conceived.
Author Response
Together with co -authors we highly appreciate you work and remarks on our manuscript. Your observations will be useful not only on this paper, but also for later studies and papers.
1.
- The methodology lacks a more detailed description of CPAP device (type, selected ventilation parameters). It would be interesting to correlate these parameters with the questionnaires scores.
CPAP devices are often used to treat sleep apnea. A CPAP device helps regulate breathing by supplying pressurized air through a hose and mask. The steady stream of oxygen helps prevent a person’s airway from becoming collapsed or blocked, which is what causes interruptions in breathing. After OSA diagnosis, each patient spent another night in the sleep laboratory for CPAP titration to select the optimal pressure at which the CPAP device eliminated abnormal breathing events.
According Spearman analysis, only Epworth sleepiness scale parameter significantly changed, depending on CPAP pressure level (p = 0,045). This point was not the main idea of our study, so it is not discussed in detail.
2.
- I highly recommend the authors to perform a ROC analysis in order to more specifically evaluate how a poor baseline score can predict the adherence to CPAP and, conversely, how CPAP can predict an improvement regarding quality of life (as expressed by questionnaires' scores).
ROC analysis is a tool to compare qualitative and quantitative data. Patients’ group who dropped out from the study (those who did not start the treatment), did not fill questionnaires after 3 months, so we do not have how to compare those subjects with those, who started treatment. After a deep discussion with our statistician, ROC analysis could not be performed.
3.
- It would be interesting to correlate the questionnaire scores with the patients’ performance at cardiopulmonary exercise testing (eg. VO2 max), which represent an increasingly used tool for quality of life assessment in CPAP patients (of course, only if some of the enrolled subjects underwent this test).
The correlation between questionnaire scores and cardiopulmonary exercise testing parameters was not made. Enrolled subjects did not undergo this test.
4.
- The small data sample represents a major drawback.
We understand small data sample issue, but this study is ongoing, so more data will be availble in the future. This issue is highlighted in limitation part of the manuscript (line 311)
5.
- The English language needs to be assessed by a native speaker, as several phrases are oddly conceived.
The manuscript was edited by English speaking editor from Edanz’s editing service (www.edanz.com). Please see the attachment “Edanz Certificate of editing”.
Thank you. Hope that our answers and comments meet you expectation.

Reviewer 2 Report
1. The abstract should mention about the statistical tests being used in the study. Also, it should be mentioned that the study is conducted with Lithuanian population.
2. All the analyses in this study are based on questionnaires. However, questionnaires are very subjective and can be biased. Since the participants already knew that they are being treated with CPAP, their responses after treatment might be heavily biased. The authors should discuss about the limitation of relying solely on questionnaires in their study.
3. The introduction is too short. It can include more details on previous studies on OSA. Is CPAP previously used before for OSA? If yes, should discuss about results that were observed. Why is this study needed? More about the motivation of this study would be beneficial.
4. In the statistical analysis, there are multiple tests performed simultaneously. All the tests should be adjusted for multiple testing correction (e.g., Benjamini-Hochberg/ Bonferroni).
In the methods, it is mentioned that a Mann Whitney U test for dependent samples is used. However, in all tables, it shows that a Wilcoxon test has been used. I assume both the tests are same. However, the authors should be consistent and clear about it for the sake of readers.
The results from Kruskal-Wallis ANOVA is not clear (line 173).
5. In line 131, the sentence is not clear.
Typo in line 153: p-value for physical component score is 0.058, which is not significant.
In line 180, the sentence is not clear.
Author Response
Together with co -authors we highly appreciate you work and remarks on our manuscript. Your observations will be useful not only on this paper, but also for later studies and papers.
- The abstract should mention about the statistical tests being used in the study. Also, it should be mentioned that the study is conducted with Lithuanian population.
- Methods part of the abstract was updated.
“Materials and Methods: Our subjects were 18–65 years old, diagnosed with moderate to severe obstructive sleep apnea and treated with CPAP between January 2020 and June 2021 in the Hospital of Lithuanian University of Health Sciences Kaunas Clinics. All patients completed the Epworth Sleepiness Scale (ESS), 36-Item Short Form Health Survey (SF-36), and Pittsburgh Sleep Quality Index (PSQI) before and after 3 months of treatment. Polysomnography was repeated as well. Statistical analyses were performed using the SPSS 27.0 software. The value of p<0.05 was considered as statistically significant. “
2. All the analyses in this study are based on questionnaires. However, questionnaires are very subjective and can be biased. Since the participants already knew that they are being treated with CPAP, their responses after treatment might be heavily biased. The authors should discuss about the limitation of relying solely on questionnaires in their study.
- The study limitations part was updated, by adding a sentence (line 311): „One more limitation of this study was that solely questionnaires were analyzed, which are very subjective and can be biased. Since subjects knew, that they were treated with CPAP devices, their responses after 3 months of treatment, could be easily predicted“.
3. The introduction is too short. It can include more details on previous studies on OSA. Is CPAP previously used before for OSA? If yes, should discuss about results that were observed. Why is this study needed? More about the motivation of this study would be beneficial.
- For patients on CPAP, it is very important to monitor care and observe the effect of treatment to reach its efficacy. Questionnaires are simple and low-effort tools to evaluate the quality of life and sleep. CPAP devices were not previously used before by these patients and this treatment method for all subjects, diagnosed with OSA, was new. This study was conducted to motivate patients to use CPAP devices, as it is efficient even after short - term of treatment with CPAP.
4. In the statistical analysis, there are multiple tests performed simultaneously. All the tests should be adjusted for multiple testing correction (e.g., Benjamini-Hochberg/ Bonferroni).
In the methods, it is mentioned that a Mann-Whitney U test for dependent samples is used. However, in all tables, it shows that a Wilcoxon test has been used. I assume both the tests are the same. However, the authors should be consistent and clear about it for the sake of the readers.
- Bonferroni test was not used in our study, due to the small sample of the subjects.
- Mann Whitney and Wilcoxon are the same tests, I could change to one therm. I agree with you. Our statistician usually uses those two terms, so maybe were there a misunderstanding.
- Line 131 sentences were updated.
“Patients that dropped out did not differ from the remaining study population in any other way. At baseline, according to chi-square and Fisher’s exact tests the pretreatment group was 14 (82.4%) men and 3 (17.6%) women. After 3 months of treatment with CPAP, the results were 15 (88.2%) men and 2 (11.8%) women (p = 1.0).”
- Line 153: value p = 0.058 is not signed as statistically significant.
- Line 173: we found significant PSQI and SF36 mental component score distribution. Mental component consists of role limitations due to emotional problems, energy (fatigue), emotional well-being, and social functioning SF36 questionnaire domains. Separately SF36 questionnaire domains and PSQI did not change.
The manuscript was edited by English speaking editor from Edanz’s editing service (www.edanz.com). Please see the attachment “Edanz Certificate of editing”.
Hope, that these clarifications meet your expectations. Thank you.

Round 2
Reviewer 1 Report
I appreciate the point-by-point answer provided by the authors and the honest manner in which they approached the reccomendations.
I consider that these preliminary results are interesting, but nevertheless need a further confirmation in a more substantial cohort of patients. Very important, a case-control study design will add certain value, even if some patients will not reach the study's final follow-up.